# Description of an Arabica Coffee Ideotype for Agroforestry Cropping Systems: A Guideline for Breeding More Resilient New Varieties

**DOI:** 10.3390/plants11162133

**Published:** 2022-08-16

**Authors:** Jean-Christophe Breitler, Hervé Etienne, Sophie Léran, Lison Marie, Benoit Bertrand

**Affiliations:** 1CIRAD, UMR DIADE, 34 398 Montpellier, France; 2DIADE, Université de Montpellier, IRD, CIRAD, 34 398 Montpellier, France; 3INECOL, Instituto de Ecología A.C., Xalapa, Veracruz 91073, Mexico

**Keywords:** *Coffea arabica* L., agroforestry, F1 hybrid, genetic resources, breeding, ideotype

## Abstract

Climate change (CC) is already impacting Arabica coffee cultivation in the intertropical zone. To deal with this situation, it is no longer possible to manage this crop using industrial agriculture techniques, which has been the main strategy implemented since the Green Revolution. Developing a more sustainable agriculture system that respects people and the environment is essential to guarantee future generations’ access to natural resources. In the case of Arabica coffee, the solution has been found. Agroforestry is proposed as an ecosystem-based strategy to mitigate and adapt to CC. At least 60% of Arabica coffee is produced in agroforestry systems (AFSs), which are the most sustainable way to produce coffee. Nevertheless, AFS coffee cultivation is currently uncompetitive partly because all modern varieties, selected for full-sun intensive cropping systems, have low yields in shaded environments. Here we review the reasons why agroforestry is part of the solution to CC, and why no breeding work has been undertaken for this cropping system. Based on the literature data, for breeding purposes we also define for the first time one possible coffee ideotype required for AFS coffee cultivation. The four main traits are: (1) productivity based on F1 hybrid vigor, tree volume and flowering intensity under shade; (2) beverage quality by using wild Ethiopian accessions as female progenitors and selecting for this criterion using specific biochemical and molecular predictors; (3) plant health to ensure good tolerance to stress, especially biotic; and (4) low fertilization to promote sustainable production. For each of these traits, numerous criteria with threshold values to be achieved per trait were identified. Through this research, an ecosystem-based breeding strategy was defined to help create new F1 hybrid varieties within the next 10 years.

## 1. Introduction

Planet Earth is in a state of decline due to human activities that have transgressed many planetary boundaries in different areas [1]. These activities and their consequences in terms of global warming result in increasing the vulnerability of agricultural and food systems [2]. In the particular case of Arabica coffee (*Coffea arabica*), the forecasted consequences of climate change (CC), and more specifically changes in rainfall patterns, elevated temperatures, more frequent drought periods, and a shift in geographical coffee-growing regions can boost environmental and socioeconomic threats in the near future [3]. *C. arabica* currently only grows in world regions where the differences in photoperiods do not exceed 2 h, suggesting that this species might not be adapted to large changes in seasonal photoperiods. A recent study provides a comprehensive view of the incidence of a short photoperiod on plant growth and transcriptomes. These results show that *C. arabica* cultivation cannot be extended beyond the intertropical zone at latitudes up to the 35th parallel [4]. Indeed, many coffee-exporting countries are also among the most vulnerable to climate risk, such as Honduras, Nicaragua, Vietnam, and Guatemala (ranking in the top 10 for climate-related damage since the 1990s) [5]. Not only coffee farmers, but actors throughout the coffee supply chain will be impacted by CC. Previous studies based on climate model scenarios predicted a 50% drop in the global area suitable for coffee production by 2050, with a drop of much as 85% in Brazil [6,7,8,9,10]. Moreover, another study predicted the extinction of a large number of wild coffee species [11]. Yet some recent publications provide some hope by showing the key role that increased atmospheric CO_2_ can play in mitigating the harmful effects of heat stress, but only if it is not associated with drought stress [12,13]. Water stress is thus likely to become the worst nightmare for coffee farmers, at least those without access to irrigation [14]. To deal with CC, it is no longer possible to use industrial agriculture techniques, which have been imposed since the Green Revolution [15]. To guarantee future generations’ access to natural resources, it is urgent to develop more sustainable agriculture systems. Sustainability is defined as the ability to continue a given behavior indefinitely [16]. For farmers this means continuing to harvest crop and animal products without degrading the resource base or the environment, while maintaining economic profitability and social stability. Sustainability encompasses processes that fulfil this goal and the ability to permanently and indefinitely maintain the required quantity and quality of resources [16]. Agricultural sustainability is based on three dimensions, i.e., environmental, social, and economic. Unfortunately, the debate is usually focused on the environmental dimension and/or the economic dimension, whereas the social dimension, including food security and human wellbeing, is often neglected [16,17].

## 2. Agroforestry Is the Best Way to Secure Coffee Production in the Future

*C. arabica* is produced in a range of different farming systems, but the two most representative ones are intensive full-sun farming systems (mechanized or not) and agroforestry systems (AFSs). Agroforestry is the collective term for sustainable land-use systems and technologies providing ecosystem services, including CC mitigation and private benefits for smallholders [18]. Biodiversity and ecosystem services are at the origin of the environment regulation and within agro-ecosystems; they are essential for sustaining future food production by contributing to natural pest control, pollination, nutrient (re)cycling, soil conservation (structure and fertility), water provision (quality and quantity), and carbon sequestration [19,20,21]. Moreover, AFS can contribute to sustainable intensification. FAO (2011) defined sustainable intensification as “producing more from the same area of land while conserving resources, reducing negative impacts on the environment and enhancing natural capital and the flow of ecosystem services”. Compared with a plot of coffee trees grown in full sun, the environmental advantages of agroforestry are evident. Coffee-based AFS, in which coffee is grown in association with other trees on the same plot of land, are widespread in many tropical countries and already represent 50 to 60% of the 8 million hectares of Arabica coffee-growing area [22]. This coffee cultivation strategy represents the best ecologically intensive agriculture model for Arabica, since trees are a way to sequester carbon, protect soil fertility, and the cornerstone of natural biodiversity. Moreover, AFSs are low-input production systems pursuant to integrated pest management (IPM) practices (Article 14 of Directive 2009/128/EC). Due to the anticipated increased future temperatures and drought periods, combined with other global CC effects, shading is likely to become an increasingly important management option and a key strategy to mitigate the negative impacts of climate variations, to stabilize production, extend tree life expectancy and conserve natural resources [23,24,25,26,27]. Tree shading is also a good solution for producing organic coffee. Shelter trees, in addition to providing additional products (fruits, wood, organic matter, etc.), are also a means to improve sub-optimal sites with poor soils and to limit water deficits, temperature fluctuations (by as much as 4–5 °C), microclimatic stress, and high wind-speed conditions [28,29,30]. Adequate shade management can also improve the water status of soil and coffee plants after prolonged drought [14]. For example, in Mexico, the annual evapotranspiration of a full-sun plantation can be estimated at 1327 mm, compared with 703 mm for crops grown under *Inga leptoloba* shade and 1052 mm for a system with a mixture of shade trees [31]. Moreover, under shade, coffee quality is generally higher, which can be explained by the higher density of coffee beans associated with more flavor intensity, notably with fine acidity and pleasant aroma [27,32]. From this viewpoint, AFS is the best way to produce specialty coffees for a constantly expanding market [33]. Economically, coffee-based AFSs aim to maintain the productivity of farming systems, reduce agricultural inputs and production costs, while diversifying production (food, fodder, lumber, building materials, and fuelwood/charcoal). AFSs also create opportunities for small-scale forest-based enterprises, while helping to reduce rural poverty by increasing on-farm production and household income. Enhancing the production diversity within farming systems can also limit the risk of economic failure [34,35]. In addition, shaded plantations should be recommended if agrochemical inputs, mechanization, and irrigation are unavailable, which is the case for many small coffee producers [28]. Socially, coffee-based AFSs, by diversifying production, can improve farmers health and nutrition. On-farm production of fuel, fodder, and other tree products saves time, money, and the effort needed to obtain them. These activities, often relegated to women, allows them to have more time for other activities. Agroforestry offers options for maximizing output per labor input, which makes it possible to reorganize tasks at the household and community levels. The perpetuation of traditional agroforestry practices can help maintain social bonds established through mutual-help arrangements [34,35]. In summary, when AFSs are properly managed, they benefit from biological, economic, and social synergies, leading to sustainable land management and higher and more stable incomes for local stakeholders (mainly smallholder farmers) [36]. Agroforestry is currently the only solution for the future of the coffee production given the CC setting because no other intensive cropping system would be viable in such a changing context. Moreover, despite all these advantages, AFS coffee cultivation is currently uncompetitive with respect to intensive systems because of their lower profitability. The world’s top two Arabica and Robusta coffee producers rely mostly on intensive systems in full sun (see https://www.visualcapitalist.com/worlds-top-coffee-producing-countries/ (accessed on 1 October 2021)). The previous lines suggest that coffee cultivation in agroforestry systems is already under way, but it is not. Over the last 20 years, landscapes as well as coffee farms across Latin America have undergone rapid and profound biophysical changes due to low coffee prices, the lack of manpower, changing climatic conditions, and severe plant pathogen outbreaks. Diverse land-use trends are affecting the sustainability of coffee-growing regions across Latin America in different ways. On the one hand Arabica cultivation is undergoing transformation from traditional to industrial production, while some coffee plantations are being converted to other agricultural land uses and coffee landscapes are also being urbanized [37].

## 3. No Coffee Varieties Have Been Specifically Selected for Agroforestry Systems

*C. arabica*, which originates from East Africa (mainly Ethiopia), was introduced in Latin America in 1720, based on a very small number of plants, which has led to a substantial founder effect [38,39]. Nevertheless, despite this initial low genetic diversity, a dwarf mutant variety named Caturra was detected and widely adopted, thereby allowing the development of full-sun intensive cropping systems, mainly in Brazil, Colombia, and Costa Rica. However, the combination of dwarf varieties, high-density cropping, and expensive pest control methods never took hold in the rest of Latin America or Africa. Although AFSs represent more than half of the Arabica cultivated area, no variety has been selected for this farming system. All modern American varieties have been bred for full-sun cultivation systems and they are not really suitable under AFS cropping conditions. This explains the drop in coffee productivity under AFSs, i.e., up to 15–80% compared with full-sun systems, depending on the shade level [27,36,40]. Why have no coffee varieties been developed yet for AFSs? There are several reasons: First, the heterogeneity of AFS makes selection more complex and this cropping system is generally used by small producers with low credit ratings. Moreover, few commercial coffee varieties are available. In the absence of seed companies, very few public or private organizations in the world have been able to develop coffee varieties (coffee breeding takes a long time, i.e., around 25 years to select a variety), and the latest breeding efforts have only been focused on crops based on Green Revolution principles [33,41]. Until rust epidemics became a huge hindrance to production, the traditional non-resistant varieties Typica and Bourbon were successfully used in AFSs. Given this situation, it seems urgent to launch breeding of new varieties adapted to agroforestry farming systems in order to cope with the major coffee-based AFS problem: its lower yields. Farmers who grow coffee in AFS are competing with those cultivating coffee in full-sun conditions with highly artificialized and unsustainable practices. They urgently need new, more resilient, and more productive varieties, which can also convince some other producers to choose agroforestry systems. This can also push some governments to promote agroforestry, as China and Vietnam have [27]. This type of genetic improvement program is part of ecosystem-based breeding [19]. Ecosystem-based breeding is ecologically oriented and aims at developing varieties adapted to ecological conditions at the regional level. In this breeding context, the new Arabica cultivars should be more in line with local pedoclimatic conditions, thereby enhancing yield stability and food security [42]. This breeding vision currently stems from the observation that production must increase without harming the environment, while ensuring agroecosystem sustainability [43]. Breeding programs can also include long-term goals, such as contributions to the maintenance or restoration of ecosystem services. For example, new varieties should be attractive for pollinators [44], contribute to the accumulation of soil organic matter [45,46], enhance soil biodiversity [47], or increase the nutrient use efficiency [48]. This “ecocentric” orientation places sustainability and ecological resilience up front, while taking the complexity of environmental factors influencing crop growth and ecosystem functioning into account. This orientation can benefit from knowledge on ecological relationships within given cultivation areas. For example, insight into local soil deficiencies should lead us to seek the best interactions of crop plants with beneficial soil organisms, such as mycorrhizas, to enhance efficient nutrient uptake [49,50], to select traits contributing to nitrogen use efficiency [48], or the best plant plasticity under CC [51]. In conclusion, for historical reasons, Arabica varieties are limited in number, with a narrow genetic base. These varieties are poorly or not at all adapted to meeting the new challenges facing coffee growing. Very few new varieties have been created and those that have been are almost always based on the same model (i.e., using the same genetic base and pedigree selection and selected for the same full-sun and intensive agronomic conditions) [41,52,53,54]. 

This can be explained by the absence of a seed industry, the small number of research institutes working on Arabica coffee breeding, the length and cost of these creations, and finally the slow diffusion and adoption of genetic advances. Regarding modern varieties, the price of field-ready plants is often heavily subsidized by governments (Brazil, Costa-Rica, Mexico) or producer federations (Colombia), while not covering the real production costs which should include research, marketing, and distribution costs. Only the newest varieties are protected, so most varieties do not have a legal owner. They are generally distributed without taking the breeders’ rights into account. Most seed producers are uncertified and are more concerned about the germination capacity than the varietal purity or plant material phytosanitary quality. There is no publicly available catalog of all varieties and regulations are seldom respected. Recently, an initiative involving several partners, including World Coffee Research (WCR), Promecafé, Borlaug Institute, Texas A&M University and USAID, caused the publication of a first partial catalog of 53 main coffee varieties (2018) (https://worldcoffeeresearch.org/work/coffee-varieties-mesoamerica-and-caribbean/ (accessed on 13 February 2018)). For all these reasons, only about twenty Arabica varieties are currently cultivated on very large surface areas, while coffee is one of the most widely traded agricultural commodities in the world. About 125 million people depend on coffee for their livelihoods in Latin America, Africa, and Asia, around 60% of which are smallholders, producing coffee on less than 5 ha. The coffee sector generates a global annual income exceeding USD 200 billion [55]. Demand for coffee is expected to grow, but the low farm profitability may hamper the supply [56]. The vast majority of smallholder coffee farmers are living below the USD 3.20 a day poverty line [55]. It thus seems relevant to launch a new genetic improvement strategy taking into account CC, Arabica genetics, the environment, and the current and future state of market needs.

## 4. Lessons Learned from the European BREEDCAFS Project

In this context, the European project Breeding Coffee for Agroforestry Systems (BREEDCAFS) (https://www.breedcafs.eu/ (accessed on 15 September 2017)) was launched in 2017. The objectives of this project were to develop, in collaboration with farmers and roasters, new breeding strategies for coffee hybrids adapted to cultivation under shade trees. The main goal was to devise new ways of tree breeding, leading to a ‘double Green Revolution’ for perennial tree crops. Using recently created *C. arabica* hybrids as a case study, BREEDCAFS wanted to show how a breeding program can benefit both smallholder farmers through new varieties for AFS with increased yields, quality, and resilience to CC, but also the European coffee industry by ensuring long-term coffee supply and the provision of an increased range of specialty Arabica coffees and modern molecular tools to assist coffee breeders.

The discovery of hybrid vigor, and the concomitant cultivation of hybrid crops represented major advances in recent agricultural history [57]. However, hybrids have seldom been created for tropical tree crops. Breaking with the traditional pattern of genealogical selection, in the early 1980s several researchers proposed to create coffee F1 hybrid varieties. The Kenyan breeding program was the first to demonstrate the advantages of hybrid cultivars, even for autogamous *C. arabica* [58]. The complex hybrid cultivar “Ruiru 11” was the first proof that it was possible to combine a compact plant type, hybrid vigor for growth and yield, high beverage quality and host resistances to coffee leaf rust (CLR) and coffee berry borer (CBD), simultaneously in the same plant [54,59]. Following this success, Arabica F1 hybrid breeding programs were subsequently adopted in Ethiopia [60], Central America [61], and Tanzania [62]. Two main factors are driving these new selection efforts: the need for high quality coffee for the specialty coffee market and the fear of a potential lack of supply. In this context, quality is paramount, alongside productivity, which implies taking genotype x environment interactions and postharvest treatments into account. Similar to many other hybrid plants, *C. arabica* F1 hybrids have genetic and agronomic advantages [57]. These hybrids have higher and more stable yields, more vigor, better disease resistance, better cup quality, and very good performance in AFS, although they have not been selected for this type of cropping system [33,63,64].

In 1990, CIRAD and partners (CATIE and National Coffee Research Institutes (ICAFEs) in Mesoamerica) began to create hybrid varieties with no emphasis on AFS or CC, using a selection process based on intercrossing of conventional American lines and ‘wild’ coffee from Ethiopia and Sudan. When cultivated in AFS, these F1 hybrids demonstrated a 30–60% yield increase without fertilizers [64], a tolerance/resistance to leaf rust and nematodes, as well as improved aromatic quality [65]. CIRAD developed the somatic embryogenesis technique to mass-propagate the F1 hybrids [66,67], a technique that was then transferred to the private sector in Nicaragua and Mexico as of 2002, which allowed the sale of around 10 million plants to farmers to date. These hybrids are studied in the BREEDCAFS program. These genetic resources providing an ideal “genepool” were studied in laboratories and tested in experimental fields and farms in Nicaragua, Costa Rica, Cameroon, and Vietnam under diverse climatic conditions. This generated new insight into coffee breeding programs and facilitated the design of future varieties (ideotype) with a better fitness to CC. Possible success in providing prediction for best F1 hybrids requires a proper definition of “the ideal ideotype from a farmer’s and roaster’s standpoint”. Arabica coffee should be regarded as a model tree crop for which most of the expected results and learning experience serves as a foundation for the improvement of other perennial tropical crops, such as cocoa, rubber, and plantain. The results obtained in this large-scale project will be widely used to define an F1 hybrid ideotype for AFS.

## 5. Select F1 Hybrids Rather Than Line Varieties

As previously mentioned, there are few existing varieties with a narrow genetic base for historical reasons [41]. These homozygous varieties are poorly or not at all adapted to the new challenges facing coffee growing. Very few new varieties have been created in the recent past and those that have been, are almost always based on the same model (i.e., same genetic base and pedigree selection). Shortening the breeding time is particularly attractive for the improvement of woody species. Trees have a long life cycle, leading to long generation times for a new cultivar with improved agronomic traits [68]. As Arabica is a self-pollinating species, the varieties developed are all relatively pure lines. The pedigree breeding method was mainly used for coffee improvement and produced most current cultivated varieties. Unfortunately, it is a long process involving different techniques (selection, hybridization, and progeny evaluation), while requiring a minimum of 25–30 years to develop a new coffee cultivar [54,69]. All major limitations associated with such conventional breeding initiatives concern the long generation time of coffee trees, the high cost of field trials, the lack of accuracy of the breeding process, differences in ploidy level between the allotetraploid *C. arabica* and other diploid species, and sexual incompatibility [41]. Moreover, it is uncertain that the objectives of varietal creation defined today will still be valid in 30 years. In the current CC setting, breeders must be able to rapidly select new varieties to avoid proposing already obsolete varieties in an uncertain future. The choice of F1 hybrid strategy reduces the time needed to create new coffee varieties by at least threefold. Further, genetic resistance to major pests such as CBD (*Hypothenemus hampei*), or to certain abiotic stresses, particularly heat and cold, are difficult to incorporate using conventional breeding techniques, notably because of a lack of knowledge regarding usable genetic resources [70]. It is therefore better to target overall stress tolerance based on F1 hybrid homeostasis and resilience. The first generation of hybrids, although selected for full-sun cultivation, was highly informative. Results showed that their yields were earlier and superior to those of American cultivars in all cropping systems tested. In terms of growth and production, the hybrids are more stable than the American cultivars across all environments, thus revealing higher homeostasis [71]. Compared with these traditional cultivars, the mean yield of hybrids was 58% higher in AFS and 34% higher in the full-sun system [64]. When properly managed, productivity can be considerably enhanced by using hybrids in coffee-based AFSs. This information can be a convincing argument to encourage coffee growers who adopted full-sun cropping systems to return to more sustainable agroforestry cropping systems. These results were confirmed in two recent studies conducted in Nicaragua [71] and Colombia [33] which also showed that F1 hybrids can exhibit excellent quality in addition to greater phenotypic plasticity [71]. F1 hybrids, by combining higher productivity, better adaptability to AFS, better resistance/tolerance, and excellent cup quality, are a promising alternative to conventional cultivars [64,71]. However, what do coffee producers think of these new varieties? A recent study assessed the perceptions of early adopters of these varieties in Central America with a view to their possible dissemination to a wide range of farmers, including smallholders. The results of interviews with farmers (ranging from micro-producers to managers of large commercial plantations in Costa Rica and Nicaragua), revealed that farmers adopt F1 hybrid varieties because they offer higher yields, fast fruiting, rapid plant growth, excellent cup quality, and fewer environmental risks than other coffee varieties. Most farmers interviewed perceive them as a safe investment. These findings show that the main barrier to adoption is the physical and financial access to the plantlets [72].

Indeed, because of their heterozygous structure, a major concern regarding F1 hybrids is that they are much more difficult to disseminate and therefore more expensive than conventional varieties. Once created, several methods are available for their large-scale multiplication, but the main difficulty to be overcome for their adoption by producers is their relatively high production costs. The first method used in East Africa consists of producing F1 seeds by hand pollination [60]. Although functional, this approach has obvious limitations when it comes to producing millions of trees. The second method is to use different micropropagation techniques to clonally produce F1 hybrid plants. Since 2006, the ECOM/CIRAD consortium has produced an average of 1 million F1 Arabica hybrids per year by somatic embryogenesis (SE) in Nicaragua [73]. Although this approach works technically, it is not economically viable due to its high costs linked to manpower and infrastructure requirements (laboratories and nurseries). Recently, the cost of F1 coffee plants was reduced by half by combining SE and rooted mini-cutting horticultural techniques [74]. Lastly, the discovery of male sterile mutants led to the development of a third method whereby F1 seeds are produced in seed gardens using a sterile male line as female plant [75]. Compared with other methods, hybrid seed production helps overcome most of the problems by having the advantage of being produced in quantity, easily handled, stored, and transported. However, the most important point is that they are much more affordable for farmers. Gaining insight into male sterility genes and their transfer should now be a research priority. F1 hybrids therefore appear to be the best alternative to meet future challenges, as proven be the case for many crop species.

Selecting F1 hybrids for AFS involves defining a specific breeding program to achieve this objective. Before defining such a program, a conceptual biological model is being designed that is expected to perform in a predictable manner in AFS. It comprises various phenotypic, physiological, biochemical, anatomical, and phenological traits which contribute to yield and quality. A modern definition of an ideotype as “an optimal combination of traits that results in an efficient matching of the plant material to its environment and cropping system, and producer and consumer demand”, should facilitate this conceptualization [76]. Ideotype breeding is an efficient method for finding solutions to various problems and developing cultivars for specific environments. Therefore, breeders need to breed varieties with traits that help the plants to adapt to CC or, in our case, to produce more in a shaded environment that buffers climate variations.

## 6. Definition of an Ideotype and Its Use in Breeding

From the outset of plant breeding, breeders have attempted to enhance yield by selecting individual traits, and this practice is ongoing. Donald (1968) added a new dimension by suggesting the breeding of model plants or ideotypes [77]. This new breeding approach was based on the premise that greater progress can be made if breeders had an ideotype as a breeding goal [78]. The ideotype approach was developed to overcome limitations of the old empirical methods of the breeders, such as “selection for yield” and “defect/default elimination”. Donald, without questioning the usefulness of these methods to create or improve varieties, proposed a more efficient alternative that starts by defining an efficient plant type theoretically followed by breeding to obtain it. According to its definition, an ideotype is a biological model which is expected to perform or behave in a predictable manner within a defined environment [77]. Guided by the idea of improving yield in cereals based on crop physiology knowledge, Donald identified key target traits for his wheat crop breeding program and succeeded in obtaining improved lodging resistance and a higher harvest index [79].

The ideotype concept is of particular interest for breeders because it reflects the idea of the best performing plant in a given environment. In practical terms, this means first identifying the main traits that should be pooled in the plant and then defining indicators to select these characteristics. Ideotype breeding is a systematic process involving various steps such as developing a conceptual plant type, selecting germplasm containing the traits, incorporating desired traits from various sources into one type, and selecting plants with an ideal combination of traits [80]. The ideotype concept can go beyond the scope of the breeding process and involve seeking the best crop phenotype to grow in specific cropping systems and for targeted end uses. This seems to be the most suitable approach with regard to coffee. The ideotype definition can therefore be broadened to include a combination of traits or their genetic bases that would confer to a variety a satisfactory adaptation to a particular environment, cropping system, and end use [78].

The ideotype approach has been expanded to include other concerns such as the market, adaptation to CC, emerging pests and diseases, and farming system changes [81]. Therefore, specific ideotypes should be designed for targeted environments. In fact, an ideotype may be designed from three different viewpoints, based on: the historical genetic vision (as described above); an agronomic view, where new genotypes are designed for a particular cropping system; and a modeling view, where the best combinations of traits are determined via formal or simulation experiments [82]. These different ideotype visions should lead to different breeding strategies. Designing an ideotype is a three-step process. First, the main objective of the selection process must be defined, then the characteristics required to meet this objective and the way to assemble them within an ideotype must be identified. Finally, a method for multi-criteria evaluation of the suggested ideotypes must be devised to determine the agronomic relevance of the integrated traits relative to the target environments [82].

If the ideotype is not viewed in terms of crop improvement through breeding but rather of adaptation to the cropping system (shade, planting density, row width, fertilization, irrigation, etc.), then the phenotypic plasticity should be exploited to obtain the desired ideotypes. Crop management is often included as a second step, thereby serving as an effective driver to complement the genetic gain, but the greatest productivity improvement may result from the selection of the ideotype in the targeted cropping context [83]. The objective is therefore to define an ideotype of a new hybrid intended for AFS with an overall organic vision of production.

## 7. Main Arabica Coffee Breeding Goals for AFS

The first ideotype breeding step is the definition of a theoretical model plant which integrates the various desired characters, and a phenotypic goal for each trait should also be specified. Then each individual trait required to achieve the phenotypic goals should be ranked in order of importance.

There are no recognized specifications for coffee-based AFSs. When the producers’ and industrial stakeholders’ expectations are taken into account, it appears that coffee productivity is paramount, but also its quality. The productivity is based on the production per tree but also tree volumes compatible with elevated crop densities that would allow high yields per hectare, while the targeted quality is determined by high sensory scores and large bean sizes. Moreover, two other traits relative to biotic and abiotic constraints seem important for coffee growing, i.e., rust tolerance and nitrogen use efficiency (NUE). This means that the ideotype must be rustic and vigorous, i.e., low fertilizer demand and high biotic stress tolerance. Since the rust fungus (*Hemileia vastatrix*) was recently able to overcome the resistance of most modern cultivated varieties, resulting in high economic losses with social implications for coffee growers [84], and based on our previous research in controlled environments [85], we believe that vigorous varieties cultivated in a shaded environment with appropriate nitrogen fertilization should be mainstreamed into an agroecological approach to control coffee diseases. In addition, most small producers cannot pay for pesticides or industrial fertilizers, and are therefore “passive organic”, without certification, because they do not have the capacity to obtain certification (time-consuming and costly). The fact that most coffee producers use little or no chemical fertilizers explains why the sensitivity to fertilization (quality and quantity) is an important trait to consider. Recently, the socioeconomic impacts of the COVID-19 pandemic in terms of labor, unemployment, stay-at-home orders, and international border policies can lead to another dramatic rust epidemic. The lack of manpower combined with a decrease in investment in farms has reduced coffee plot maintenance, in turn creating conditions that can be conducive to future shocks [86]. As an ideotype specifies the ideal attributes of a plant for a particular purpose, in Table 1 we propose a list of traits in order of decreasing importance under a pre-defined crop management strategy and cropping system.

## 8. Identification of Morpho-Physiological Traits to Achieve a Specific Goal

### 8.1. Trait 1: Productivity

#### 8.1.1. Vigor and Tree Volume

Tree vigor can be considered as an important contributing factor in productivity and earliness. Due to vigor, particularly hybrid vigor, varieties and hybrids present differences in vegetative volume. A major yield component is determined by the crop density, which is directly dependent on the canopy volume [33]. In Arabica, all dwarf modern varieties have the same single gene that confers dwarfism. The widespread presence of the ‘Caturra’ dwarfism gene found in the 1950s in Brazil [87] led to a reduction in tree volume in modern American varieties, thereby enabling cultivation densities of 4500 to 6000 trees/ha, compared with only 2500 to 3000 with tall conventional accessions. One of the pillars of the Green Revolution is the reduction in plant volume and subsequently in cultivation intensification [88]. Arabica coffee is no exception to this rule. Dwarfism has also had substantial benefits in fruit tree production, enabling higher yields, and facilitating harvesting in orchards in full-sun conditions, as well as in AFSs [89]. In the case of Arabica, productivity per plant has not been increased, on the contrary, but the reduction in tree volume has enabled increased productivity per hectare [33]. Similarly, Moncada et al. (2019) showed that the yield potential of any pure line variety cultivated in Colombia today is lower than that of various wild Ethiopian accessions, thus confirming that modern breeding has not increased the productivity per plant but rather has increased the productivity by means of increased density [90]. On the other hand, F1 hybrids, which inherit the dominant dwarfism gene, can individually produce much more while maintaining the advantages of high planting density. Hybrid vigor is the only way to significantly and rapidly increase productivity per tree. Trunk diameter is the simplest and earliest indirect measure of hybrid vigor. The second indirect measure is the number of growth units (internodes) established per unit of time. This measure is particularly interesting as it reflects the performance under abiotic stress (i.e., drought and high temperatures). The second morphological trait linked to productivity is tree volume, with at least 4000 trees cultivated per hectare. New varieties must therefore have a volume of between 0.5 and 1.0 m^3^ to allow a high planting density [33].

Early vigor may be measured by the stem diameter of 1-year-old plants in the field or by the increase in stem diameter between the first and second year. In Côte d’Ivoire, *C. canephora* displayed a genetic coefficient of correlation between vigor and productivity ranging from 0.70 to 0.93 [69].

Trunk diameter measurements made on very young Arabica hybrid trees after 3 months of field cultivation can predict future vigor [91]. This vigor is correlated with the performance index (PI) value, which is another key indicator of productivity and plant health [85,91]. PI measures the performance of photosystem II (PSII) and the efficiencies of specific electron transport reactions in the thylakoid membrane [92]. The demonstration that chlorophyll fluorescence measurements can be used to estimate the operating quantum efficiency of electron transport in coffee leaves, and its direct relation to coffee plant health and oxidative stress level has led to the use of chlorophyll fluorescence for examining photosynthetic performance in diverse situations [85,93]. By using this approach, in this study we demonstrated that the information encoded in fluorescence transients can be efficiently employed in high-throughput screening for growth and vigor under open field conditions. Fluorescence data were used instead of molecular markers and phenomic prediction-based models were built [91]. Their predictive abilities with respect to leaf counts, tree height, and trunk diameter were evaluated and the findings revealed that phenomic prediction with fluorescence data outperformed genomic prediction under all environmental conditions. First, the physiological traits linked to productivity are vigor and photosynthesis performance, which can also contribute to earliness and plant health.

#### 8.1.2. Floral Induction

Most actors involved in coffee production (farmers, researchers, agronomists, etc.) are unanimous in saying that productivity per tree is lower in agroforestry systems [27,40], yet no one knows exactly why from a physiological viewpoint. Moreover, very few scientific articles address this key issue with regard to AFS profitability [94,95,96]. Before setting productivity improvement targets, the physiological basis for this yield decrease, probably associated with shading, must be determined. The fundamental question that should be answered to be able to accurately determine the targeted traits is: “Why do coffee trees, which are understory trees, produce much less when they are cultivated in AFS”?

This question is hard to answer because there is no single type of agroforestry practice [27]. In addition to the fact that the generic term AFS may apply to very different situations, with shading intensity ranging from 5% to 80%, we may be talking about an agroforestry system set up with trees selected for their wood quality, citrus or banana plantations, a mixture of these trees, or with heterogeneous forest undergrowth (Figure 1). Moreover, it is technically difficult to maintain both homogeneous and constant shading.

The most common argument is that, similarly to most crops, all modern coffee varieties have been selected for full-sun cultivation. This argument does not answer the question. In 2004, DaMatta addressed this issue in a review where he identified three main causes of this production loss. First, lower whole-tree carbon assimilation under excessive shading can explain the decline in coffee yields. It may also be due to a greater stimulus for vegetative rather than flower bud growth. Finally, coffee yield can decrease if fewer nodes are formed per branch and flower buds at existing nodes [36].

We addressed this issue in the framework of the BREEDCAFS project. Some hypotheses were tested by studying the Marsellesa variety, i.e., a modern American pure line variety (Sarchimor) which is known to be not very productive and unsuitable for AFS. Five hypotheses were tested to determine the origin of the yield loss in AFS: (1) insufficient photosynthesis, (2) a change in plant architecture (plasticity), (3) an improper conversion of flowers into fruit, (4) low flowering, (5) or a combination of these assumptions.

The Marsellesa variety was studied on three plots, each with a share of agroforestry and full-sun conditions at three elevations. The production data monitored over 2 years (2018–2019) showed a drastic decrease in productivity under natural shade compared with full-sun conditions (Figure 2). The conventional coffee biennial production cycle was clearly represented, and during both high and low production years, production in full sun was always at least 25% higher. The extent of production loss under shade was 25–80%, depending on the location, elevation, and year. In 2018, production was reduced by 61%, 54%, and 80%, respectively, at 800, 1100, and 1300 m.a.s.l. The 2019 harvest was slightly better under shade, with losses of 35, 26, and 68%, respectively, at the same elevations. Plots at lower and higher elevations always provide the highest losses under AFS, with the worst productivity noted at 1300 m elevation. The tree volume, size, collar diameter, branch size, internode length, and leaf surface and density were measured to determine whether natural shade had a significant impact on plant architecture [96]. The volume of Marsellesa trees was similar under full sun or high natural shade, regardless of the elevation. The same results were obtained under artificial shade [33]. Leaves were significantly larger under shade at the two lowest elevations. Leaf density remained stable regardless of the elevation or growing conditions. Regarding the tree size, collar diameter, branch size and internode size, there were no observable differences between full-sun and shade conditions at all elevations. Overall, none of the measured parameters varied significantly when comparing natural shade and full-sun conditions, so we therefore were unable to explain the yield differences measured under the two light conditions.

A portable spectrophotometer was used to select plots with similar natural shading at the three elevations by measuring the light spectrum quality and intensity received by plants. This revealed that under strong natural shade, the average peak light wavelengths ranged from 476 nm under full sun shift to 778 nm. The light intensity was 17-fold lower in the agroforestry conditions used. Finally, the wavelength analysis showed the red (660 nm)/far-red (730 nm) photon flux ratio (Pr/Pfr) were completely different under natural shading, i.e., around 0.5 in AFS versus 1.2 in full-sun conditions. These measurements also highlighted differences between strong natural shading and artificial shading. The latter simply reduced the light intensity without changing the light spectrum quality. We previously established a close correlation between chlorophyll a fluorescence and net photosynthesis (unpublished data). Chlorophyll a fluorescence and net photosynthesis measurements for all six growing conditions showed that photosynthesis was more effective under shade, and that plants were less stressed. Surprisingly, the low light intensity received in agroforestry conditions and the modification of the light spectrum perceived by coffee trees did not affect the photosynthesis [94]. Lower whole-tree carbon assimilation did not seem to be the cause of the yield decrease under natural shade.

Productivity is comparable when Marsellesa is grown under full sun, regardless of the elevation. Yet this productivity declines under shade, with the lowest productivity obtained at high elevation. We compared the percentage of branches bearing fruit under both conditions, and the percentage of flowers converted into fruit to gain insight into why the Marsellesa cultivar produces less under shade (Figure 3). The percentage of branches bearing coffee cherries was particularly stable across the cropping conditions and elevations. The percentage of flowers converted into fruits was found to not be the factor determining the loss of production under shade, although it varied substantially depending on the elevation. At 800 m, this percentage was similar under both light conditions and reached only 30%. At the medium elevation, almost 50% of the flowers were converted into fruits under full sun and 70% under shade. At 1300 m, the conversion was similar under full sun (50%) but lower under shade (33%) (Figure 3). These results highlight the influence of temperature on this flower–fruit conversion. The high temperatures encountered at 800 m had, for example, the same negative impact in full sun and under shade. High temperatures reduced flowering while inducing floral malformations which led to their abortion [97]. Flowering was always higher in full-sun conditions at 1100 and 1300 m and similar under shade and full sun at low elevation (800 m) in both conditions. (Figure 3).

Surprisingly, this problem of floral initiation under shade has been completely overlooked in coffee research. Highlighting this issue opens new perspectives for research to better understand the underlying physiological mechanisms in Arabica, and to develop methodological solutions or tests to assess the potential of different genotypes with a view to creating varieties adapted to shading.

### 8.2. Trait 2: Quality

Of the large volumes of coffee produced under AFS, an ever-increasing quantity is being sold as “specialty coffee.” There is renewed interest from roasters and the market in the fact that coffee varieties are key factors with regard to coffee quality [98]. With the emergence of the specialty coffee market, breeders now focus on genotypes with specific aroma and fragrance, such as Geisha with fruity and zesty notes and sometimes described as having floral, jasmine, and peach-like aromas or Ethiopian profiles with flowery notes [99]. Specialty coffee is defined by the Association of Fine Coffees (SCA, https://sca.coffee/ (accessed on 1 September 2021)), based on both the product quality and the quality of the livelihoods of everyone involved in this coffee cultivation [100]. The SCA protocol considers that coffees with a score above 80 fall into the specialty coffee category (scale from 1 to 100). Specialty coffees are distinguished from mainstream coffees on the basis of several criteria, including quality, sustainability, and closer relationships with growers [101]. The coffee sector is becoming increasingly aware of the need to increase both productivity and quality. Unfortunately, the profitability of farms that produce so-called specialty coffees is not included in the list of criteria. Although the terroir and elevation are generally taken into account, specialty coffee roasters have begun to understand that the choice of genotype has a huge impact on sensory quality. Wild or old varieties with high organoleptic qualities, such as Geisha, Typica, or Bourbon, are becoming keys to the market success of the highest quality specialty coffees [102]. Nevertheless, these old varieties that produce the best coffees are also relatively unproductive and very susceptible to coffee rust and other diseases. They are only profitable if the producers often manage to find buyers willing to pay them a fair price [103]. Nevertheless, the exceptional quality of a very small number of varieties is recognized. One of the questions to be addressed is: “Could new varieties be bred for the specialty market that combine exceptional quality and good productivity in AFS”? New indicators of quality useful for specialty coffee breeding, but also for characterizing the physiological and biochemical traits linked to quality, are beginning to be identified [71,104].

Criteria based on the physical characteristics of green coffee beans, are still first-rate at the time of purchase of green coffee. The bean size is an important quality criterion for roasters. For example, only large-size beans are marketable for specialty coffee, because bigger beans generally produce coffee with more aroma. Nevertheless, smaller beans of the same variety are classified as being of lower grade and in turn their prices are lower [105]. On the international market, the green coffee bean size and defect number determine the bean quality before roasting; the higher the percentage of grains between 16 and 20, the better the quality. The 100-grain weight (W100) is widely used for other crops [106]. In some studies, this coffee bean measurement appears to be representative of the bean density and therefore the seed filling quality during the fruit development and ripening phases [105,107]. There is a significant relationship between the W100 and the final sensory score. W100 appears to be a good cup quality predictor. This parameter is simpler to apply than the green bean size parameter in assessments [71].

Other studies have focused on two major green coffee compounds that are considered to be responsible for poor sensory quality (chlorogenic acids) and good sensory quality (lipids). By comparing five genotypes of Arabica (Geisha especial, Geisha 3, ET47, Marsellesa, and T5175), at two elevations (700 and 1300 m) and two development stages (the last two, when the fruit is yellow and then red), Marie et al. (2022) showed that the measurement of the lipid and chlorogenic acid content of grains can serve as a quite easy-to-use quality indicator [104]. Both genotypes are also acknowledged to be of good (Geisha especial) or on the contrary poor (T5175) sensory quality. For example, the Geisha especial genotype contains more lipids than the T5175 genotype, and with lower chlorogenic acid content at 1300 m elevation. Moreover, using a transcriptomic and metabolomic approach, the authors highlighted strong influences of genotype x environment interactions on the terpene pathway. Identification of limonene synthase, i.e., a monoterpene synthase which promotes limonene accumulation in green and roasted coffee beans, allows differentiation of Arabica genotypes in their environment and can be a reliable predictor of quality [104].

Although currently not widely used, near infrared spectrophotometry (NIRS) can revolutionize quality assessment methods. NIRS is a rapid and low-cost technique that can be applied to the analysis of green or roasted beans (coffee in grain or powder forms). The main problem comes from the calibration curves or equations that must be developed before this technique can be effectively used. NIRS is now used to predict the total content of the main family of precursors: lipids, chlorogenic acids, sugars, trigonelline, and caffeine [65,108]. NIRS technology is already effective for quality prediction in roasted coffee [109]. However, further work is needed to improve the technique and develop calibration curves for quality precursors in green coffee with the aim of using it in the future in phenomic selection for beverage quality. While awaiting the tailoring of NIRS for coffee assessment, a recent study demonstrated the possibility of using HPLC in a simple and efficient way to evaluate the quality of commercial coffee samples [110].

### 8.3. Trait 3: Plant Health

Global warming is already a major threat to coffee trees, causing severe epidemics of coffee leaf rust (CLR) in many coffee producing countries in South and Central America, resulting in high economic losses with social implications for coffee growers [84,111,112,113]. The situation is the same from Mexico to Brazil. Resistance genes introgressed from the Timor hybrid no longer offer total resistance to new rust races. This is becoming a major concern in the coffee sector because all known SH major resistance genes have already been exploited in breeding [114]. Developing healthy plants that are able to defend themselves through horizontal resistance mechanisms, while being productive even when attacked by the pathogen, should be part of a more rapid and sustainable alternative approach [84,115,116].

It was recently shown that an agroecological approach to disease control can be at least partly based on vigorous varieties combined with a shaded system and appropriate nitrogen fertilization [85]. In other terms, if vertical resistance is overcome, a sufficient dose of partial resistance may remain and depend on the plant physiological status, as demonstrated by Echeverria-Beirute et al. (2018) who studied the relationship between leaf/fruit balance and rust incidence [117].

The plant physiological status and particularly the level of oxidative stress, is closely correlated with rust sensitivity [85]. In this context, hybrid vigor can contribute to reducing the disease incidence. Among the many parameters studied, including mineral elements, sugars, hormones, and secondary metabolites, those related to photosystem II and photosynthetic electron transport chain components only appeared to be indicators of the physiological status of coffee plants and able to predict rust sporulation intensity. These physiological parameters can be easily estimated by measuring chlorophyll a fluorescence, which makes it possible to calculate a performance index (PI) [92]. Secondly, the same authors demonstrated that hybrids have an increased photosynthetic electron transport efficiency, better carbon partitioning, and higher chlorophyll content, as already demonstrated in several plants such as Arabidopsis [118], rice [119], and maize [120]. The chlorophyll content measured on plants cultivated in phytotrons or in the field (full-sun and shade) was always higher in hybrids compared with line varieties and/or their parents [93]. The same authors also demonstrated a close relationship between the expression of genes related to the photosynthetic electron transport chain and chlorophyll a fluorescence. Taken together, these results led to the definition of three new indicators of productivity and rust tolerance: PI, chlorophyll content, and the oxidative stress level. These indicators are simple, non-destructive, inexpensive, and rapid measurement tools to assess the coffee tree physiological state. Rust tolerance (or resilience) depends on the physiological status of the coffee plant, which itself depends on the plant health at the moment of the rust penetration and adaptation potential (homeostasis) to the environment and agronomic conditions.

Several recent studies have highlighted the relationship between rust incidence and productivity [117,121]. Surprisingly, it was shown that the genotype selection index (GSI), which takes both productivity and productivity stability into account, was not related to the rust incidence level [71]. Susceptibility to rust cannot be predicted by a high or low GSI, and moreover being sensitive or not to rust can lead to a high GSI. Hybrid vigor actually reflects a better photosynthetic efficiency and better carbon partitioning, which can ensure a better productivity–adaptability trade-off.

### 8.4. Trait 4: Low Fertilization

As cropping systems, particularly coffee, require careful nitrogen (N) management, increasing the nitrogen use efficiency (NUE) is a way to achieve enhanced sustainability. An overall improvement in NUE requires an increase in its nitrogen uptake, nitrogen use efficiency and nitrogen harvest index components. Of course, each of these components involves numerous crop physiological mechanisms and agronomic characteristics, which explains the NUE complexity [48]. Nitrogen supply, more than that of any other nutrient, is the most limiting nutrient for coffee production, particularly during the vegetative growth period [122]. Any deficiency during this short period is known to have lasting effects on subsequent coffee bean production [123]. Coffee fields planted at densities below 5000 trees/ha require 150 to 250 kg N/ha annually [123,124]. The amount of nitrogen fertilizer to use is dependent on both the density and the cropping system. For example, in agroforestry systems, shade tree litterfall alone can provide up to 95 kg N/ha/year [125]. Trials have shown that some varieties, especially hybrids and wild accessions, are able to develop properly when nitrogen levels are low. The findings also helped identify interesting accessions for breeding programs. In the framework of the BREEDCAFS project, greenhouse facilities have been used to monitor the NUE of various *C. arabica* genotypes: wild accessions (Rume Sudan, E531, sterile male), lines (Marsellesa, Caturra, T5296, Iapar59), and F1 hybrids (H1, H3, Marianna and Starmaya). Two nitrogen fertilizer levels were used to fertilize the plants for 6 months. The results showed that all the tested genotypes had better photosynthetic parameters under a high nitrogen supply (N+) compared with a low nitrogen supply (N-). However, under the low N conditions, some of them had a higher PI than others, such as the H1 hybrid. This hybrid presented a high final biomass that was equivalent under both nitrogen levels. PI is also a reliable indicator of the ability of varieties to use soil nitrogen. A second experiment was conducted in tunnels in Costa Rica, with wild *C. arabica* accessions. Plants were supplied with two different N fertilizers for 6 months, and morphologic, phenotypic, and photosynthetic parameters were recorded during and at the end of the experiment. Some wild accessions (i.e., E344, E340, ET5, etc.) had high biomasses under both nitrogen levels. These accessions are therefore good candidates for breeding programs targeting low nitrogen farming systems (S. Léran, pers. Com.)

Moreover, as mineral N fertilizer production is based on fossil energy [126], fertilizers are becoming increasingly expensive. Since September 2008, if we look at the average price of each nutrient collected in the Illinois cost of production dataset, the following increases may be noted; anhydrous ammonia (+118%), urea (+101%), liquid nitrogen (+84%), DAP (+50%), MAP (+61%), and potash (+61%) (https://www.fb.org/market-intel/too-many-to-count-factors-driving-fertilizer-prices-higher-and-higher (accessed on 13 December 2021)). However, breeding for NUE has not yet been extensively mainstreamed into practical breeding programs for many crops. NUE is a complex trait to assess and it is also very much influenced by the soil conditions, so it is difficult to control. In terms of breeding, at the rooting system level, the objective would be to improve the N uptake efficiency [48]. Breeding under low nitrogen conditions would allow the selection of plants with the best N uptake and root distribution over the soil profile [127]. Is it possible to select F1 hybrids with early and flexible root formation, high root depth, biomass, and length density?

At the cropping system level, the goal is to design an optimal soil and nutrient management system so that N is kept in the system while keeping in mind that each harvest corresponds to a net loss. Therefore, strategies are needed to replenish soil nitrogen reserves, such as no-till, continuous cover cropping, as well as crop residue and soil organic matter preservation, while maintaining crop yields and profit margins for farmers. Nitrogen-fixing trees and legumes can play an especially important role as they are able to fix free atmospheric N2 [128]. Recently, there has been increasing attention on microbial ecology in the rhizosphere which may facilitate nutrient availability and uptake [129]. The role of crop below-ground traits, such as the root architecture and symbiosis with soil microorganisms, are being specifically studied with regard to N-fixing bacteria and mycorrhizas that support N uptake [48]. Such agronomic approaches to improve NUE are challenging as they are subject to strong genotype x environment (GxE) interactions [130].

Since it is possible to adapt cropping system management strategies to improve NUE, it is not only the GxE relationships that must be taken into account but also the whole cropping system, and NUE must be considered as a result of genotype, environment and management (G×E×M) interactions [131]. Each management strategy impacts NUE, and applying different types of fertilization management (e.g., low or high N input, organic or conventional fertilizers) requires different agronomic management strategies to improve NUE while also likely requiring adapted genetic traits [132,133].

## 9. Coffea Arabica Ideotype for AFS Cropping Systems

Based on results and available information in the morpho-physiological traits presented above, we propose the following coffee ideotype for agroforestry cropping systems (Table 2).

## 10. Possible Ways to Select Targeted Traits within an Ideotype

By using traits responsible for high yields in real cropping conditions, precisely measurable and earliness during the tree selection process, breeders are therefore able to select target traits earlier [80]. Indeed, the identification of measurable traits involved in the adaptation of the plant to specific cropping conditions would help to improve the characterization of varieties, to unravel the GxE interaction observed in variety trials, to define the ideotype best adapted to local cropping conditions, and finally to identify ideotype-like varieties among the available possibilities.

Genetic diversity among the wild coffee individuals is sufficient to warrant this breeding effort [39]. Diversity is available in open access Arabica germplasm collections, such as the CATIE collection in Costa Rica [38,134] or in national collections of wild and landrace accessions in Ethiopia [135]. F1 hybrids, combining higher productivity, better adaptability to AFS, better resistance/tolerance to biotic and abiotic stress, and higher cup quality, are a promising alternative to conventional cultivars [115,136]. Moreover, a range of new F1 hybrid varieties may be selected within the next 10 years due to the somatic embryogenesis multiplication technique, associated with the horticultural rooted mini-cuttings method [73,74]. As mentioned, different natural sterile males are also available to produce F1 hybrids that can be multiplied by seed. Hybrid varieties will therefore be provided preference for cultivation in agroforestry systems, with the possibility of crossing line varieties between them, line varieties with wild accessions (F1 hybrids), F1 hybrids between them (double hybrids), or with line varieties or wild accessions (three-way hybrids). Because of the difficulty of conducting several breeding cycles to increase the likelihood of the trait contributing to higher yield, larger-scale breeding programs should be developed to make effective use of the vast genetic diversity available. The first step will consist of pre-selecting families of new hybrids under 50% shade, then selecting the best individuals within each family before multiplying them, and finally conducting multilocation clonal trials. The following is an example of how to do this. After 100 crosses have been performed, the 100 families of 20 plants obtained will be placed in a nursery for 6 months (Figure 4). Vigor-related and plant health-related traits will be measured at this time. Although water stress is not a priority in agroforestry, a tolerance test involving the measurement of the fraction of transpirable soil water (FTSW) will be performed [137]. Root length measurement can also be considered prior to transplanting the plants in the field. By applying a selection index of 80, the 20 least tolerant and vigorous families will be eliminated. Eight hundred plants, or 40 families, will be planted in an agroforestry field with approximately 50% shade. At the end of the first year, measurements of the tree collar diameter, size, volume, and chlorophyll content will be used to select the 32 most vigorous families (selection index of 80). At the end of the second year, the flowering earliness will be measured for each family. Finally, over 3 years of production, productivity and quality will be evaluated each year under low nitrogen fertilization conditions. At the end of this selection process, between 10 and 12 families will be pre-selected (selection index of 40) and multiplied before the multilocation trials.

## 11. Conclusions

The genetic improvement program described above will be implemented in 2023. The pre-selection stage will involve two sets of crosses, each aimed at producing 50 hybrid families, which will be staggered by 1 year. In 2029, about 20 hybrid families will be pre-selected and multiplied in order to carry out multilocation tests. In 2035, the best hybrids created for the first time for agroforestry systems will be ready for dissemination.

Breeding programs have so far been focused on above-ground plant parts. Coffee, as well as other fruit tree crops, is perennial and must therefore cope with soil constraints due to CC. The underground plant compartment, which has been the focus of very little research, should be taken into greater consideration in breeding. The BOLERO project (Breeding for coffee and cocoa root resilience in low-input farming systems based on improved rootstock, HORIZON-CL6-2021-BIODIV-01-13), which was recently accepted by the European Union, will start in 2022 and proposes to boost knowledge on the root system architecture to develop breeding programs dedicated to root plasticity and adaptation. Traits such as deep rooting, branching, and root system plasticity in various stress conditions will be specifically studied. BOLERO aims to decode the coffee and cocoa root microbiome through a microbial metagenomics approach to determine the impact of coffee/cocoa genotypes, different farming systems, environment (e.g., N levels, drought, pesticide levels, geographic location) on the root microbiome, and its encoded life-supporting functions. The project proposes to create resilient rootstock varieties for fruit tree crops to cope with CC threats. A fast and cheap rootstock breeding strategy is the main goal. A precise root ideotype definition is also essential to build this new breeding program. Integrating high-throughput root phenotyping, with microbiome profiling, metabolic modelling, and genomic selection, will help the coffee sector produce varieties adapted to sustainable agronomic systems and easily accessible via open-source seed systems. The time and costs of perennial crop selection can be reduced by specifically breeding for root and shoot parts, associated with efficient and reliable micropropagation techniques such as micro-grafting.

## Figures and Tables

**Figure 1 plants-11-02133-f001:**
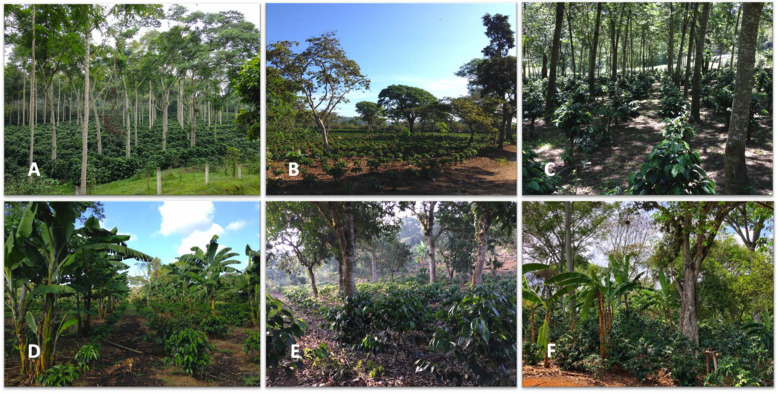
Examples of coffee farming in agroforestry systems. (**A**) AFS set up with a single shade tree species (Veracruz, Mexico). (**B**) AFS with a low level of shade and a mixture of tree species (Veracruz, Mexico). (**C**) AFS with a high level of shade set up with a single shade tree species (Matagalpa, Nicaragua). (**D**) AFS set up with banana and citrus fruit trees (Veracruz, Mexico). (**E**) AFS set up in heterogeneous forest undergrowth with a low level of shade (Veracruz, Mexico). (**F**) AFS set up in heterogeneous forest undergrowth with a high level of shade (Veracruz, Mexico). Photos © Jean-Christophe Breitler.

**Figure 2 plants-11-02133-f002:**
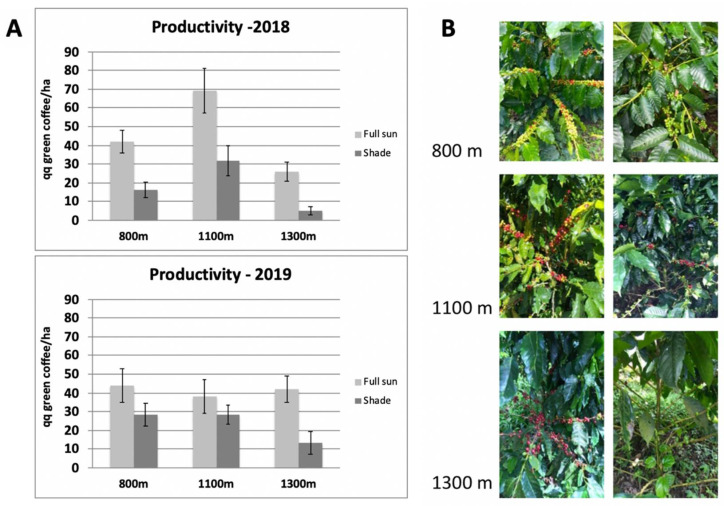
Biennial variations in production depending on the elevation and shading of the Arabica cv. Marsellesa. (**A**) Yield comparison of trees grown in full sun and under shade at three elevations. We studied three plots in Nicaragua planted with the modern variety Marsellesa. Marsellesa plants were planted in three crop fields in 2014 at three different elevations in the “La Cumplida” farm (Matagalpa, Nicaragua). The “Santa Fe” field is located at 800 m elevation (12°59′59″ N, 85°52′8″ O), the “El Coyol” field is located at 1100 m elevation (13°1′5″ N, 85°51′6″ O) and the “El Panorama” field is located at 1300 m elevation (13°1′1″ N, 85°50′32″ O). Each field is divided into two parts, one of which is in full sun all day. The second corresponds to an agroforestry system with a high shading level of above 50%. Yield was measured in quintals of green coffee per hectare. Yield was estimated over two growing seasons (2018—2019). This assessment was based on 25 trees per condition, counting the number of cherries per six branches at three heights representative of the tree. The average was then multiplied by the number of productive branches to obtain the number of cherries per tree. In order to assess the standard error of this method, the exact number of cherries was counted for several trees. The standard error was thus estimated at +/− 10%. The same evaluation method was used to estimate the number of flowers per tree. (**B**) Representative pictures of fruit load of trees grown in full sun or under shade at three elevations.

**Figure 3 plants-11-02133-f003:**
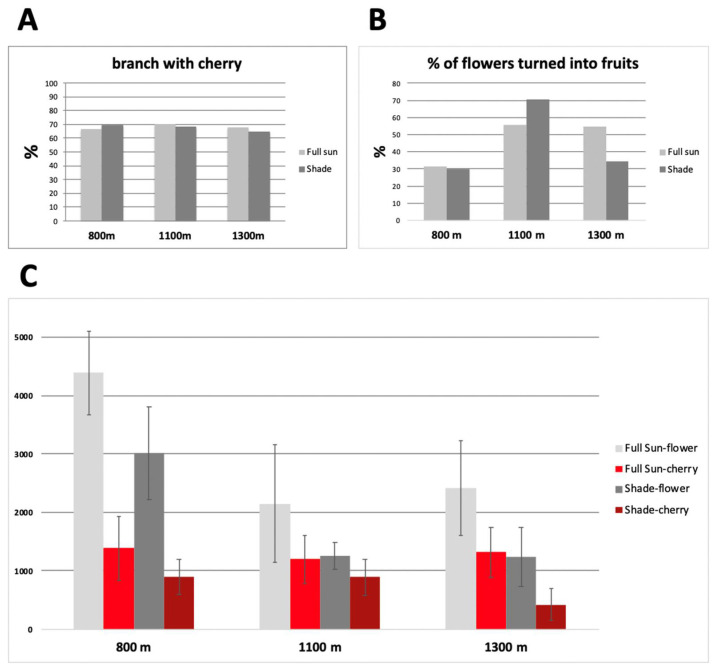
Flowering and productivity measurements of Marsellesa cv. cultivated in full sun and under shade at three elevations. (**A**) Comparison, expressed as a percentage, of the number of fruit-bearing branches in the two growing conditions at the three elevations. (**B**) Percentage of flowers converted to fruit. (**C**) Number of flowers and fruits counted from 25 trees for each condition and elevation.

**Figure 4 plants-11-02133-f004:**
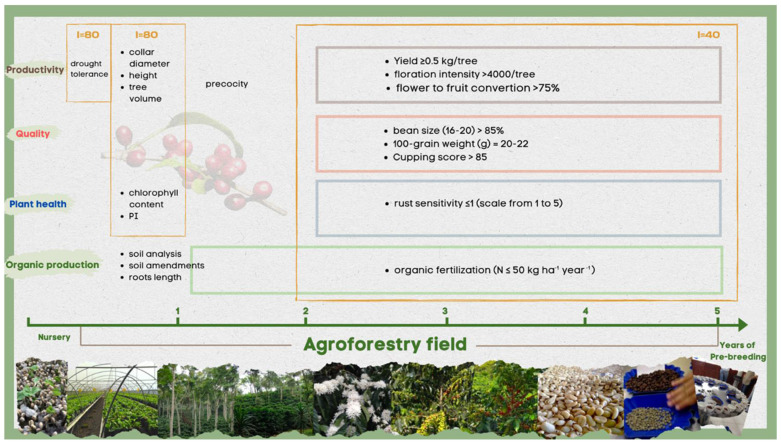
Methodology for pre-selection of new Arabica hybrid families. Over 5 years, the hybrid families will be evaluated first in the nursery, and then in the agroforestry field (50% shade) by applying different selection indices based on the criteria defined as the desired ideotype. (i = selection index).

**Table 1 plants-11-02133-t001:** Objectives in passive organic agroforestry cropping systems with a list of principal traits that are to be part of the coffee ideotype breeding initiative with a phenotypic goal for each of them. (* 100 grain-weight, ** Performance indices).

Cropping System		Agroforestry
Crop management		Organic or «passive organic»
Trait ranking	Traits	Phenotypic goal based on
Trait 1	Productivity	yield (green coffee kg/plant)
Trait 2	Quality	Cupping score, bean size, W100 *
Trait 3	Plant health	vigor, high PI **
Trait 4	Low fertilization	NUE, P and K

**Table 2 plants-11-02133-t002:** Proposed ideotype for 15 traits of coffee hybrids to be grown in agroforestry systems.

Individual Traits	Phenotypic Goal
Productivity	
Yield (Kg of green coffee/plant)	>0.5
Floriation intensity	>4000 flowers/plant
Flowers to fruit conversion	>75%
Tree volume (m^3^)	0.5–1
Collar diameter (6 months old)	18–20 mm
Crop density	4000–5000 plants/ha
Plant height	180–250 cm
Precocity	2 years
Quality	
Bean size (16–20)	>85%
100-grain weight (g)	20–22
Cupping score	>85
Plant health	
Chlorophyll content (mg/gFW)	>2
Performance Index	highest
Rust sensitivity	<1(sporulating lesions reaching 1 to 5% of the total leaf area [110])
Organic production	
Nitrogen use efficiency (NUE)	25–50 kg N ha^−1^ annually

## Data Availability

Not applicable.

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
