# Peer review of "Description of an Arabica Coffee Ideotype for Agroforestry Cropping Systems: A Guideline for Breeding More Resilient New Varieties"

_plants, 2022, doi:10.3390/plants11162133_

Round 1
Reviewer 1 Report
In the bibliography the Author should also consider the following articles, recently published in Natural Product Research.
Quality assessment of Coffea arabica commercial samples.
Agnieszka Viapiana, Filippo Maggi, Mateusz Kaszuba, Pawel Konieczynski & Marek Wesolowski.
Natural Product Research, Volume 34, 2020 - Issue 21
Author Response
The reviewer's recommendation was taken into account and the reference was added to the review. A paragraph corresponding to this publication has been added at the end of section 8.2.
Reviewer 2 Report
. The authors Jean-Christophe Breitlier et.,al have done a commendable job in their manuscript entitled “ Description of Arabica coffee ideotype for agroforestry cropping systems: a guideline for breeding more resilient new varieties.’’The authors have described how climate change place a very important role in coffee cultivation. The authors also highlight the agroforestry system as a solution for climate change. The authors also describe four major traits such as (1) productivity based on F1 hybrid vigor, tree volume and its flowering intensity under shade; (2) beverage quality by using wild Ethiopian accessions as female progenitors and selecting for this criterion using specific biochemical and molecular predictors (3) plant health to ensure good tolerance to stresses, especially biotic; (4) low fertilization to promote sustainable production. All the characteristics identified above could help in more directed and specific breeding techniques in the years to come. The authors towards the end have described through the pictures coffee farming in agroforestry which aids as a perfect visual aid. Their analysis on the productivity clearly shows the dependence of the coffee cultivation on altitude and shading. A similar analysis has also been carried out on flowering and productivity measurements in full Sun and under shade at three different altitudes. Finally, the authors described a methodology for pre-selection of new Arabica hybrid families. Overall, I find this manuscript well-written, informative and extremely comprehensive.
Minor comments
1) I would suggest the authors to revamp their writing style of asking questions under sections 2,3,4,5,6,7 and instead refer to such paragraph titles as statements.
2) I would also request the authors to make the description crisp and drive home their points through more concise and compact and small descriptions rather than lengthy.
3) Finally, I would also request the authors to have the manuscript checked by native English speaker and run through the manuscript for any grammatical or sentence construction errors.
Author Response
-The following section titles 2,3,5,6,7 have been modified as recommended by the reviewer
-The manuscript has been fully proofread by a professional translator: David Manley - Translator/Editor
d.manley@numericable.fr
Reviewer 3 Report
General comments
Please check the MDPI format on references!!
Please revise the article based on the MDPI format!!!
Please answer my questions:
What is the objective of this study?
What is the novelty of this study?
Why did you use your last articles to cite?
Specific comments
Line 50 a large number of change to many!
Line 56 which has been remove it
Line 61-64 revise, please
Line 82 and represent already change to already represent
Line 110-111 revise please
Line 114-116 revise please
Line 160-162 revise please
Line 166-168 revise please
Line 194 the newest ones are protected; therefore, most varieties do not have a legal owner
Line 279 other species that are diploid change to other diploid species,
Line 289 although selected for cultivation in full sun, hybrids were rich in information
Line 356-361 revise please
Line 421-423 revise please
Line 530-535 revise please
Line 618-621 revise please
please revise the article base on the English Editor native.
Author Response
What is the objective of this study?
The objective of this review is to make an inventory of coffee growing, to define the current and future threats and to define selection objectives for future varieties. The definition of an ideotype for agroforestry gives the objectives to be reached for the creation of new F1 hybrids adapted to this type of cropping system.
What is the novelty of this study?
No one has ever selected coffee varieties specifically for agroforestry. We explain why, and we explain why we think this is so important today. This review is the result of our work to prepare this selection work, in particular by using the concept of ideotype which seems to us very important to both set clear and precise objectives and to avoid forgetting important criteria.
Why did you use your last articles to cite?
A large part of the definition of this coffee ideotype for agroforestry stems from our recent work, notably in the framework of the European project Breedcafs. That is why we quote our publications. This selection program is the result of more than twenty years of work within our team.
Specific comments
Line 50 a large number of change to many!
done
Line 56 which has been remove it
Line 61-64 revise, please
done
Line 82 and represent already change to already represent
done
Line 110-111 revise please
done
Line 114-116 revise please
done
Line 160-162 revise please
done
Line 166-168 revise please
done
Line 194 the newest ones are protected; therefore, most varieties do not have a legal owner
done
Line 279 other species that are diploid change to other diploid species,
done
Line 289 although selected for cultivation in full sun, hybrids were rich in information
done
Line 356-361 revise please
done
Line 421-423 revise please
done
Line 530-535 revise please
done
Line 618-621 revise please
done
The manuscript has been fully proofread by a professional translator: David Manley - Translator/Editor
d.manley@numericable.fr

Round 2
Reviewer 3 Report
The manuscript has been revised according to the reviewer's comments. Thank you for the responses.
best regards